# Immunotherapy in Breast Cancer: Beyond Immune Checkpoint Inhibitors

**DOI:** 10.3390/ijms26083920

**Published:** 2025-04-21

**Authors:** Yeonjoo Choi, Jiayi Tan, David Lin, Jin Sun Lee, Yuan Yuan

**Affiliations:** Division of Medical Oncology, Department of Medicine, Cedars-Sinai Medical Center, 8700 Beverly Boulevard, Los Angeles, CA 90048, USA; yeonjoo.choi@cshs.org (Y.C.); jiayi.tan@cshs.org (J.T.); david.lin@cshs.org (D.L.); jinsun.bitar@cshs.org (J.S.L.)

**Keywords:** breast cancer, immunotherapy, bispecific antibody, cell-based therapy, cancer vaccine, oncolytic virus

## Abstract

The systemic treatment of breast cancer has evolved remarkably over the past decades. With the introduction of immune checkpoint inhibitors (ICIs), clinical outcomes for solid tumor malignancies have significantly improved. However, in breast cancer, the indication for ICIs is currently limited to triple-negative breast cancer (TNBC) only. In high-risk luminal B hormone receptor-positive (HR+) breast cancer (BC) and HER2-positive (HER2+) BC, modest efficacy of ICI and chemotherapy combinations were identified in the neoadjuvant setting. To address the unmet need, several novel immunotherapy strategies are being tested in ongoing clinical trials as summarized in the current review: bispecific antibodies, chimeric antigen receptor T-cell therapy (CAR-T), T-cell receptors (TCRs), tumor-infiltrating lymphocytes (TILs), tumor vaccines, and oncolytic virus therapy.

## 1. Introduction

The treatment landscape of breast cancer (BC) has been considerably transformed with the introduction of multiple targeted therapies that resulted in a significant improvement in patient survival [1,2]. In HR+ BC, CDK 4/6 inhibitor, mTOR inhibitor, and PI3K/AKT inhibitor have expanded therapeutic options for HR+ BC and delayed the initiation of chemotherapy [3,4,5,6,7,8]. Human epidermal growth factor receptor 2 (HER2)-targeted therapy is the mainstream for HER2-positive (HER2+) BC. In addition to conventional dual HER2-targeted monoclonal antibodies, HER2 tyrosine kinase inhibitors (TKIs), antibody–drug conjugates (ADCs) such as trastuzumab emtansine (T-DM1) or trastuzumab–deruxtecan (T-DXd) have significantly improved clinical outcomes of HER2+ BC [9,10,11]. Moreover, poly-adenosine diphosphate–ribose polymerase inhibitors (PARPis) have shown promising clinical benefits in both high-risk early-stage BC and advanced BC, in patients with germline BRCA 1/2 mutation [12,13,14]. In addition, ADC targeting trophoblast cell surface antigen 2 (TROP2) or HER2-low and HER2-ultra-low BC also showed improved outcomes in HR+ and TNBC [15,16]. Lastly, emerging data from immune checkpoint inhibitors (ICIs) in TNBC have changed the landscape of treatment for TNBC [17].

Among all breast cancer subtypes, TNBC is considered the most immunogenic, with higher tumor-infiltrating lymphocyte (TIL) and programmed death-ligand 1 (PD-L1) levels [18,19]. In the KEYNOTE 355 trial for metastatic TNBC, the addition of pembrolizumab to chemotherapy demonstrated modest progression-free survival (PFS) and overall survival (OS) benefits in PD-L1-positive mTNBC [17]. In early-stage TNBC, the incorporation of ICIs with neoadjuvant chemotherapy has significantly improved pathological complete response (pCR), a surrogate marker for survival. Pembrolizumab in combination with chemotherapy received FDA approval for TNBC based on the KEYNOTE522 trial, with a pCR of 64.8% vs. 51.2% with chemotherapy alone [20]. The estimated OS at 5 years was 86.6% in the pembrolizumab–chemotherapy group, as compared with 81.7% in the placebo–chemotherapy group (*p* = 0.002) [21]. A non-anthracycline-containing regimen (NEOPACT) using carboplatin docetaxel and pembrolizumab for six cycles showed a comparative pCR rate of 58% with an encouraging toxicity profile [22]. The ongoing SWOG phase III SCARLET trial is evaluating the NEOPACT regimen in comparison with the KEYNOTE522 regimen (NCT05929768) [20,22]. The utilization of ICIs is currently limited to PD-L1+ TNBC in the metastatic setting, which accounts for 25–40% of the mTNBC population. This underscores substantial unmet needs in the novel immunotherapy strategies for breast cancer.

This review aims to provide a comprehensive overview of immunotherapy modalities beyond ICIs in breast cancer.

## 2. Immunotherapies in Breast Cancer

### 2.1. Bispecific T-Cell Engagers and Bispecific Antibodies

Dual-specific protein targeting is a promising approach in developmental immunotherapy. Bispecific T-cell engagers (BiTEs) consist of two parts—one binds to tumor-specific antigen (TSA), while the other binds to the CD3 receptor expressed on T cells, and upon binding, T cells release cytotoxic molecules such as perforin and granzymes. Aside from their direct killing effects, BiTEs can enhance the proliferation of T cells even at low concentrations and modulate the tumor microenvironment—by recruiting antigen-presenting cells (APCs)—to promote favorable outcomes [23]. Several efforts are underway to investigate the role of BiTEs in breast cancer treatment [24]. Surface proteins upregulated in BC such as HER2, epithelial cell adhesion molecule (EpCAM), or TROP2 are common targets [25,26]. Ertumaxomab, a BiTE targeting HER2 and CD3, has exhibited antitumor effects in HER2-low tumors refractory to trastuzumab [27]. In the phase I trial, dose-limiting toxicity (DLT) was not reached and one of fourteen patients achieved a partial response [28,29]. A BiTE targeting P-cadherin and CD3 (PF-06671008) showed preclinical activity in mouse models [30]. However, the phase I trial in humans did not show efficacy, with significant treatment-related adverse events (TRAEs); the most common was cytokine release syndrome (CRS), leading to permanently discontinued treatment in 25% of the patients (NCT 02659631) [25]. Preclinical data on bispecific antibody targeting Trop2 and CEACAM5 demonstrated antitumor efficacy in xenograft models [31,32,33].

Other than BiTEs, there are several bispecific antibodies (BsAbs) which bind at least two epitopes of TSA instead of T-cell markers. Lymphocyte activation gene 3 (LAG3), a protein that acts as an inhibitory receptor, usually in exhausted T cells, is considered a promising target for immunotherapy [34]. Tebotelimab, a BsAbs targeting PD-1 and LAG-3, was evaluated in a phase I trial and showed a 6% response rate in the TNBC cohort (*n* = 31) (NCT03219268) [35]. In HER2+ BC, the combination of tebotelimab and margetuximab achieved an overall response rate (ORR) of 19% (14/72) [35]. There were 5 confirmed responses among the 28 PD-L1^−^ patients who had disease progression on prior anti-HER2 therapy (18% ORR). B7-H3 is frequently overexpressed in breast cancer and modulates the proliferation and activation of T cells [36]. Over 30 clinical trials targeting B7-H3 through a variety of mechanisms are conducted: bispecific antibodies, ADCs, CAR-T, CAR-NK, and radioimmunotherapy [37,38]. B7-H3 and CD3 BsAbs showed inhibitory effects on the xenograft model [39]. The combination of B7-H3/CD3 BsAbs with ICIs is ongoing (NCT03406949).

Ivonescimab is a BsAb that targets both PD-1 and vascular endothelial growth factor (VEGF). Ouyang et al. reported its excellent antitumor activity and a manageable safety profile (NCT05227664) [40,41]. In a phase I/II study (*n* = 36), patients with mTNBC received ivonescimab with paclitaxel or nab-paclitaxel as the first line, and a PFS of 9.4 months was achieved. Interestingly, ORR reached 80% regardless of PD-L1 status. In the patients with PD-L1 combined positive score (CPS) < 10% group, ORR was 79%. All patients (*n* = 36) experienced a treatment-emergent adverse event (TRAE) of any grade; 18 (50.0%) patients experienced a TRAE of grade 3 or higher, and 9 (25.0%) experienced serious TRAEs. The most common TRAEs of grade 3 or higher were neutropenia at 19.4%, and increased liver enzyme at 5.6%. Another BsAb, PM8002/BNT327 targeting PD-L1 and VEGF-A, also demonstrated promising clinical benefit when combined with nab-paclitaxel in a phase Ib/II study (NCT05918133, *n* = 42) [42]. PFS was 13.3 months, and the ORR was 73.8%. More strikingly, the ORR was 77%, and the median PFS was 18 months in patients with CPS < 1%. Over half of patients experienced grade 3 or 4 TRAEs (54.8%). A phase III randomized trial is planned for verification of these findings. Both agents showed remarkable efficacies in PD-L1-negative mTNBC, which could potentially become practice-changing. Clinical trials with BiTEs in breast cancer are described in Table 1.

### 2.2. Cancer Vaccines

Cancer vaccines can be classified into several types by antigen-recognizing platforms: peptide-, protein-, carbohydrate antigen-, tumor cell-, DNA-, and dendritic cell (DC)-based vaccines [43]. The most common type of cancer vaccine is peptide-based, which usually consists of 20–30 amino acids derived from tumor-associated antigens (TAAs). It can be taken up by DCs and presented on major histocompatibility complex (MHC) class I molecules to form the peptide–MHC-I complex [44], allowing it to be recognized by cytotoxic T cells and ultimately leading to the elimination of cancer cells. HER2 is considered a potential target for peptide-based vaccines. Despite early success in inducing systemic immune responses and having excellent safety profiles, NeuVax^TM^ in adjuvant settings failed to show significant benefit in phase III trials [45]. In the phase III trial (NCT01479244) of NeuVax^TM^, a total of 758 patients with operable low HER2 BC were enrolled, and there was no significant difference in disease-free survival (DFS) between the control and experimental arms (*p* = 0.069) [45]. GP2 is a nucleopeptide derived from HER2. A recent phase II trial for patients with residual disease or high-risk pCR after receiving neoadjuvant or postoperative anti-HER2 agents reported that the GP2 vaccine had no significant benefit in DFS in the total population, yet a subset analysis showed a trend toward improved DFS in the HER2+ group that received the GP2 vaccine (*p* = 0.052) [46]. A phase III trial of GLSI 100 (GP2 vaccine plus granulocyte–macrophage colony-stimulating factor (GM-CSF)) will be launched (NCT05232916) [47].

Human telomerase reverse transcriptase (hTERT) is a subunit of the telomerase complex that facilitates cell proliferation. It is known to be related to unfavorable prognosis in several cancers [48,49]. A vaccine targeting hTERT (VX-001) was tested in human leukocyte antigen (HLA)-A2+ patients with MBC (*n* = 19). No objective response was observed and 50% of patients had stable disease (SD) [50]. Tissue analysis did reveal increased TILs and tumor necrosis, indicating augmented intratumoral immune responses [50]. Additional trials are ongoing for further investigation of this vaccine in MBC (NNCT00573495 and NCT01660529).

A whole protein-based vaccine which presents TAAs was investigated in HER2+ BC. Compared to peptide-based vaccines, it contains both HLA class I and II epitopes, allowing it to overcome HLA restriction. In a phase I study, a vaccine targeting intracellular domains of HER2 (amino acids 676 to 1255 of the full-length HER-2/neu) demonstrated the immunogenic profile in 29 HER2+ breast and ovarian cancer patients. A total of 89% of patients developed T cell-mediated immunity against HER2 and 82% had an increased level of HER2-specific immunoglobulin G antibodies [51]. A phase I study (NCT00058526) evaluated the efficacy of recombinant HER2 protein (dHER2) combined with the AS15 immunostimulant [52]. In this study, patients with stage II-III HER2+ BC were enrolled. After 5 years of follow-up, 62% of patients were disease-free, and dHER2-specific antibody responses were induced, with the rate of responders increasing with the dHER2 dose and the number and frequency of immunizations. The study was limited by its single-arm nature without vaccine-free control.

Glycans are the most common surface antigens presented by cancer cells, and glycosylation plays a pivotal role in cancer development and progression. Therefore, various efforts targeting glycans as an anticancer vaccine were made over the years, resulting in many clinical trials that have proven unsuccessful, largely due to the immune tolerance associated with them [53]. Novel strategies such as the selection of appropriate domains or antigen modification were used to enhance carbohydrate-based cancer vaccines [54]. Sialyl-Tn (STn) is a carbohydrate antigen that is related to poor prognosis in solid tumors. The vaccine STn-KLH (Theratope^®^) was the first carbohydrate-based vaccine exhibiting efficacy in a phase II trial [55]. Based on this success, a randomized, double-blinded, phase III trial was conducted and 1028 patients with metastatic BC received cyclophosphamide with this vaccine [56]. Overall, the treatment was well tolerated, and considerable antigen-specific immunoglobulin generation was observed. However, the vaccine failed to improve PFS and OS [57].

The DNA-based vaccine uses DNA fragments that encode TAAs. It is delivered by plasmids or viral vectors and engulfed by APCs, which are subsequently recognized by T cells [58]. The selection of targeting antigens and the formulation of the optimal delivery approaches remain challenges for DNA-based vaccines. One approach targets the transmembrane glycoprotein mucin1 (MUC-1), which is overexpressed on cancer cells and promotes cancer cell immigration and invasion by activating the transforming growth factor beta (TGF-β)-related signaling pathway [59]. PANVAC is a vaccine with carcinoembryonic antigen (CEA) and MUC-1 DNA fragments encapsulated in viral vectors [60]. A phase II trial study tested PANVAC with/without docetaxel in patients with MBC (*n* = 48). Median PFS (mPFS) was numerically better in the PANVAC arm, but not statistically significant (7.9 vs. 3.9 months, *p* = 0.09) [61]. Mammaglobin-A (MAM-A) is a TAA overexpressed in 40–80% breast cancers and is known to affect tumor growth by changing chemotherapy sensitivity [62]. A phase I trial was conducted to evaluate the safety and efficacy of the DNA-based vaccine targeting MAM-A [63,64]. HLA-A2 or 3 were initially selected based on preclinical findings (*n* = 8), and patients without HLA-A2 were also enrolled later on (*n* = 6), and no statistically significant improvement in PFS was found [65,66,67].

HER-2 DNA vaccine with GM-CSF and interleukin-2 (IL-2) was tested in a Swedish trial, and increased immune activity was observed [68]. A phase I trial testing a DNA vaccine co-targeting HER2 and CEA was conducted in patients with II-IV BC, and no definite cell-mediated immune response was observed [69]. A DNA vaccine targeting HER2 ICD was tested in a phase I dose-escalation trial in patients with stage III and IV HER2+ BC (*n* = 66). Most patients experienced grade 1 or 2 adverse events, and the patients receiving higher doses of vaccine were correlated with a higher magnitude of HER2 IDC type I immune responses (NCT00436254) [70]. At the time of the report, the median OS and PFS had not been reached in any arm [70]. Another phase I trial utilizing a xenogeneic HER2 DNA-based vaccine for patients with advanced-stage BC is currently ongoing (NCT00393783).

A DC-based cancer vaccine is also under development. DC-based vaccines are produced in ex vivo environments, allowing them to adopt an individual’s immune characteristics by being exposed to various TAAs [71]. This individualized processing is a technical hurdle and makes it challenging to assess the response [72]. In a phase 1 trial including ductal carcinoma in situ (DCIS) and early-stage HER2+ BC, variable immune responses in peripheral blood and tumor specimens were observed (66.7% to 89.5%). Patients with DCIS had a higher pCR than HER2+ BC patients (28.6% vs. 8.3%) [73]. In addition to HER2, p53-derived peptide DC vaccine, with co-simulation by IL-4 and GM-CSF, was tested in patients with metastatic BC (*n* = 26). Eight patients had stable disease, with a positive correlation between response and p53-specific cellular immune response [74].

Using tumor cells is one of the classic concepts in cancer vaccine development. The intact tumor cells can induce humoral and cellular immunity by themselves, and the antigens derived from the cells can also induce substantial immune reactions [75]. However, this can often lead to off-target immune responses, including autoimmune diseases, because of their allogeneic source [76]. A combination vaccine with autologous DCs and tumor cells was evaluated in a phase I trial [77], in which patients (*n* = 32) with metastatic breast or renal cancer were enrolled and modest efficacy was observed, with 1 patient achieving near complete response (CR) and 1 patient having partial response (PR) in the breast cancer cohort (*n* = 16). BriaCell is a human immortalized cell line, engineered to trigger both adaptive and innate responses [78]. In heavily pretreated patients with mBC, a phase 1/2 study (NCT03066947) showed that low-dose cyclophosphamide followed by the SV-BR-1-GM regimen plus retifanlimab, then low-dose local pegylated interferon alpha (IFN)-α, demonstrated improved PFS and OS with acceptable tolerability, and the disease control rate (DCR) reached 50% [79]. A sequential phase 3 trial displayed higher median overall survival (OS) (13.4), an objective response rate (ORR) of 9.5%, and a clinical benefit rate (CBR) of 55% (NCT06072612) [80]. Another phase I/II trial of a V-BR-1-GM regimen in combination with retifanlimab is ongoing in patients with metastatic or locally recurrent breast cancer who have failed standard therapy (NCT03328026) [81]. Completed and ongoing trials are described in Table 2.

Messenger RNA (mRNA) vaccination’s success in combating coronavirus disease (COVID-19) has propelled recent interest in mRNA vaccination in breast cancer [82]. mRNA vaccination has the potential to convert immune-cold breast cancer into immune-hot breast cancer and synergizes with immune checkpoint inhibitors. Effective mRNA vaccine design for in vivo function requires consideration of many factors, including effective vaccine targets, mRNA structures, transport vectors, and injection routes. In a recent phase 1 trial, intratumoral immunotherapy with pembrolizumab and mRNA-2752, a lipid nanoparticle-encapsulated mRNA encoding three immunomodulatory proteins, OX40L, interleukin (IL)-23, and IL-36γ, was tested in patients with ductal carcinoma in situ (DCIS) [83]. The combination was associated with flulike symptoms and grade 1 or 2 local reactions with no serious systemic toxic effects and resulted in an 80% overall response rate. All patients with ERBB2-positive or hormone receptor-negative DCIS experienced at least a partial response, with one-third experiencing complete resolution of the tumor, and high-baseline immune cells were associated with response.

### 2.3. Adoptive T-Cell Therapy

Adoptive T-cell therapy is one of the immunotherapy strategies that utilizes infusion of the manufactured T cells to eliminate cancer cells [84]. Manufactured T cells can be derived from autologous or allogenic sources and then modified to increase immunogenicity and antigen recognition.

#### 2.3.1. CAR-T Therapy

Chimeric antigen receptor T cells (CAR-T) utilize T cells with antigen recognition domains specific to TAAs and immune-boosting signaling domains. CAR-T has been widely used in routine clinical care for patients with hematologic malignancies, with significant benefits in extending PFS and OS [84]. Its adaptation in solid tumors has been challenging, which is largely attributed to tumor heterogeneity and the hostile tumor microenvironment. There are several target molecules used for CAR-T therapy: HER2, Mesothelin, c-Met, ROR-1, and NKG2D.

HER2 overexpression in breast cancer renders it a highly actionable cancer target. Early HER2 CAR-T development encountered a variable challenge. An early trial reported a patient developed fatal acute respiratory failure after lymphodepleting chemotherapy and infusion of a high dose of HER2 CART designed with the trastuzumab single-chain variable fragment (scFv) and high-dose interleukin (IL)-2 [85,86]. Additional trials administered a lower dose of HER2-specific CAR-T designed from non-trastuzumab scFvs after no or low-intensity lymphodepletion and without cytokine support; although toxicity was not observed in the regimen, antitumor activity was limited [87,88,89]. Other Her2 CAR-T trials such as NCT02713984 and NCT04650451 were terminated due to safety concerns. Currently, a phase 1 trial of CCT 303–406 is anticipated to recruit patients with metastatic HER2-positive solid tumors (NCT04511871). At the same time, HER2 CAR-monocytes/macrophages (HER2 CAR-M) became an appealing option due to the enrichment of macrophages/monocytes in the TME, with potentially better tumor penetration. HER2 CAR-M CT-0508 showed modest antitumor activity and a tolerable safety profile in patients in a phase 1 study (NCT04660929). Among patients who received CT-0508 (*n* = 14), SD was found in 28.6% of patients [90,91,92].

Mesothelin is overexpressed in most TNBC (~67%); therefore, several clinical trials aimed to evaluate mesothelin-targeted CAR-T products for TNBC. Wang et al. generated PD-1 and T-cell receptor (TCR)-deficient CAR-T products and conducted a phase 1 trial (NCT03545815). In this trial, one mTNBC patient was enrolled, showing 20% disease regression after receiving two doses [93]. Several new early-phase trials are recruiting patients now (NCT05623488, etc.).

C-Met is a member of the MET family, and disrupting the C-Met signaling pathway leads to the promotion of malignant cells. C-Met expressions in breast cancer varied from 14% to 54% and showed higher expression in TNBC than in other subtypes [94]. A few CAR-T trials were conducted for solid tumor expression of C-Met protein. A phase 0 trial in intratumoral injection of mRNA-transfected c-Met-CAR T cells (NCT01837602) aimed to assess safety and feasibility in mTNBC. Among six patients, CAR T messenger ribonucleic acid (mRNA) was detectable in the peripheral blood of two patients and the injected tumor tissues of four patients. The injections were well tolerated, without drug-related adverse effects greater than grade 1 [95]. Shah et al. performed a phase 1 trial of autologous C-Met CAR-T in metastatic melanoma and TNBC patients (NCT03060356) [96]. Among patients who expressed the protein more than 30% by IHC, four patients with TNBC and three patients with melanoma were treated with the CAR-T product. All patients experienced Gr 1-2 AEs, and half of the TNBC cohort achieved SD. Additionally, mRNA response was detected in all patients [97,98,99].

Receptor Tyrosine Kinase-like Orphan Receptor 1 (ROR-1) is involved in embryonal development [100], and it is highly expressed in breast cancer and related to its growth [101]. An anti-ROR-1 CAR-T trial (NCT02706392) showed that two of four TNBC patients reached SD after 15 weeks and 19 weeks of treatment and that none of the four patients experienced severe neurotoxicity or CRS [102]. A phase 1 trial of another ROR-1-targeted CAR-T treatment (LYL797) is currently recruiting solid tumor patients (NCT05274451) [103].

NKG2D (natural-killer group 2, member D) plays a role in the immune activation of NK cells and T-cell subsets by recognizing tumor cells [104]. The first-generation NKG2D-targeted CAR-T therapy showed an antitumor effect in in vivo experiments with the MDA-MB-231 TNBC cell line [105]. Therapeutic Immunotherapy with NKR-2 (THINK) is a multinational open-label phase I study that will evaluate the safety and efficacy of autologous NKR-2 CAR-T cells (NCT03018405). In addition, NCT04107142 will evaluate the access feasibility and safety of haplo/allogeneic NKG2DL-targeting CAR-T cells in solid tumors. In addition to these trials, there are a number of CAR-T trials in BC (Table 3).

#### 2.3.2. TCR Gene Therapy

T-cell receptor gene therapy is another type of adoptive cell therapy. An engineered T-cell receptor is expected to overcome the limitations of CAR-T by increasing the presentation of membrane and intracellular proteins and thus increasing MHC processing [106]. Therefore, TCR gene therapy is more sensitive than CAR-T even in lower-density epitopes. NY-ESO-1 (New York esophageal squamous cell carcinoma 1) is a cancer–testis antigen and is presented in multiple types of solid tumors [107]. Despite its potential for immunotherapy, several TCR-based therapies were conducted in early-phase clinical trials without success. Its effectiveness in humans has been limited due to several potential causes: heterogeneous antigen expression leading to incomplete tumor eradication; limited persistence of TCR-T cells; tumor immune escape mechanism through downregulation of HLA-A molecules; or loss of tumor NY-ESO-1 expression [108,109].

CEA is another antigen that is expressed in less than half of breast cancer cases [110], while elevated serum level is also associated with mBC [111]. Autologous T lymphocytes genetically engineered to express a murine T-cell receptor (TCR) CEA were tested in a phase I trial and the results for the first three patients were reported. All patients experienced profound decreases in serum CEA levels (74–99%), and one patient had an objective regression of cancer metastatic to the lung and liver. However, severe transient inflammatory colitis that represented a dose-limiting toxicity was found in all three patients, which led to the termination of the trial [112]. A phase 1 trial of CEA-targeted therapy (NCT00673829) aimed to address safety and efficacy in mBC patients, but the trial was suspended in 2016 with no result reported.

#### 2.3.3. Tumor-Infiltrating Lymphocyte Therapy

Tumor-infiltrating lymphocyte (TIL) therapy in an intratumoral environment is considered a key factor in antitumor immunity [113]. Due to prior activation, TILs isolated from patients’ tumor tissues demonstrate enhanced recognition of target antigens compared to naive T cells. The first drug for TIL therapy, lifileucel (Amtinagvi), was approved for melanoma, and its indication has expanded to other solid tumors. A pilot phase II trial (NCT 01174121) demonstrated a promising result for BC; six patients received autologous TIL with pembrolizumab, and ORR was 50%, including one case of complete response and two partial responses [114]. The trials for TNBC are currently recruiting more patients (NCT04842812 and NCT04111510).

### 2.4. Oncolytic Virus

Oncolytic virus is an emerging immunotherapy that induces antitumor immunity indirectly and oncolysis directly [115,116]. Talimogene laherparepvec (TVEC) is based on the herpes simplex virus (HSV) and is genetically engineered to penetrate cancer cells. A phase 1 trial included 14 patients with MBC among a total of 30 patients with solid tumors, where 13 patients received a single dose, and 17 patients received multiple doses without DLT. A total of 19 of 26 patients with post-treatment biopsies had residual diseases, with 14 having tumor necrosis [117,118,119]. Furthermore, a phase II trial with neoadjuvant TVEC therapy in early TNBC was conducted (NCT02779855) [120]. In the study, the RCB0 rate was 45.9% and RCB0-1 was 65% without DLT. Not all results were promising, as seen in a phase II trial (NCT02658812) which explored intratumoral administration of TVEC for inoperable breast cancer; the result seemed disappointing. There was no completion of the planned treatment. Seven of nine terminated the study due to rapid deterioration, and SD was the best response in the remaining two patients [121]. Similarly, when TVEC was combined with atezolizumab in a phase 1 trial (NCT03256344), only 1 patient had PR among 11 patients with TNBC [122]. However, another phase 2 trial of TVEC injection followed by stereotactic body irradiation and pembrolizumab in mTNBC (*n* = 28) presented a different result [123]. Patients tolerated the treatment well, two patients reached CR, and one patient reached PR with a durable response. Additionally, a phase 1 trial for HER2-negative, metastatic, or unresectable disease (NCT03554044) demonstrated a safety profile and local disease control near the injection site [124].

LTX 315 is an oncolytic peptide which showed a potential demolishing effect when combined with doxorubicin or ICI. According to the preclinical data, LTX 315 and doxorubicin resulted in tumor regression and local necrosis in TNBC mice models [125]. A phase 1 study of LTX 315 with pembrolizumab or ipilimumab included 17 patients with TNBC. The concurrent pembrolizumab group showed a PR rate of 17% among patients with evaluable disease (NCT01986426) [126].

Pelareorep is derived from human reovirus. A phase 2 BRACELET-1 study (NCT04215146) evaluated the efficacy of pelareorep in patients with metastatic HR+/HER2− disease [127]. Patients who received pelareorep plus paclitaxel showed a median PFS of 12.1 m, compared to 6.4 m in the paclitaxel-alone arm, with a hazard ratio of 0.39. In the AWARE-1 trial (NCT04102618) (*n* = 38), pelareoprep plus atezolizumab was compared to pelareoprop monotherapy [128], and TIL was increased in the atezolizumab arm. Another phase 1 study of pelareoprep with paclitaxel in metastatic HR+/HER2 patients (*n* = 15) showed an ORR of 20% [129].

Oncolytic virus based on an adenovirus platform has a long history of development. Onyx-015 was the first drug to enter clinical trials and showed limited efficacy in mBC [130]. To overcome the limitations, deletion of E1ACR2 was introduced to increase clinical activity. A phase 1 trial (ICOVIR-7) gathered a total of 21 patients with metastatic solid tumors, 3 of whom had mBC. All patients showed manageable toxicity, and one patient with BC obtained a stabilized tumor marker [131].

Other viruses like pox, polio, or vaccinia have been utilized to develop oncolytic peptides. Early-phase clinical trials are active or recruiting populations with solid tumors including TNBC (NCT03564782, NCT02630368, NCT04301011, and NCT02432963). The novel chimeric orthopoxvirus, CF33-hNIS-anti-PD-L1 (CHECKvacc), encodes two transgenes, human sodium–iodide symporter (hNIS) and anti-PD-L1, with robust anticancer activity in TNBC xenografts. TNBC cells infected with this virus express functional hNIS and anti-PD-L1 proteins. CHECKvacc intratumoral injection is currently undergoing investigation in patients with mTNBC as a single agent (NCT05081492) [132].

### 2.5. Cytokine Genes

Cytokines induce inflammatory responses among cells and ultimately lead to tumor death [133]. Recombinant interferon-α and interleukin (IL)-2 were approved for the treatment of some malignancies such as melanoma. NKTR-214, also known as bempegaldesleukin (BEMPEG), is an engineered IL-2-specific agonist that stimulates the expansion and proliferation of CD8 T cells and NK cells. NKTR-214 has been tested in a phase 1 trial (NCT02869295) in advanced or metastatic solid tumors (*n* = 28 total; *n* = 2 BC) and proved its clinical efficacy, with tolerable toxicity and evidence of increased CD4 and CD8 T cells in the tumor microenvironment (TME) [134].

IL-12 is a proinflammatory cytokine that increases the production of interferon-γ (IFN-γ), and has generated interest in its combination with immunotherapy. In an in vivo study with a BC cell line, tumor shrinkage was observed when recombinant adenovirus encoding IL-12 was administered [135]. A phase 1 trial was initiated to evaluate therapeutic potential as a monotherapy in TNBC, and increased TIL was observed in locoregional TME [136]. Moreover, cytokine therapy has expanded its indication when combined with other immunotherapies. The phase 2 study KEYNOTE 890 (NCT03567720) was aimed at assessing the efficacy of intratumoral tavokinogene telseplasmid, a plasmid encoding IL-12, followed by electroporation and pembrolizumab in patients with mTNBC. Eleven patients completed response evaluation after treatment, three (27.3%) achieved PR, and evidence of enhanced immune response related to IL-12 was detected in blood [137].

### 2.6. Immune-Modulating Agents That Target the Innate Immune System

SD-101 is an intratumoral toll-like receptor 9 (TLR9) agonist. Its mechanism is to increase the production of IL-2, which leads to cytotoxic activity in TME [138]. The combination of SD101 and the known KEYNOTE regimen was studied in I-spy2 (NCT01042379). Despite showing an enhanced achievement rate of pCR in patients with high-risk, HR-positive/HER2-negative stage II/III breast cancer (*n* = 75), the regimen did not meet the pre-specified threshold for graduation [139].

Rintatolimod is a chemokine modulator used to treat myalgic myeloencephalitis/chronic fatigue syndrome. It acts on TLR-3 and enhances the innate immune system. A phase 2 trial was completed in an mTNBC cohort in combination with IFN-2a and celecoxib (NCT03599453), and the result from a phase 1 trial for early TNBC announced that toxicity profiles were manageable and the rate of pCR plus ypT1mic was 66% [140].

## 3. Conclusions

Despite the emergence of immune checkpoint inhibitors, there are several limitations that ICIs cannot fulfill in the treatment of breast cancer. They are closely related to the original pathologic mechanisms of BC and its tumor-favorable microenvironment. Modulating those circumstances is the key to advancing immunotherapy, as several classes of drugs were introduced to manage the limitations. As well as addressing TME, the effective recognition of target antigens is another key to drug development. In addition to HER2, several proteins have proved their potential. Moreover, improving binding power and infiltration of recognized T cells is an important challenge. A newer class of antibodies, vaccines, adoptive T-cell therapy, etc., provides ways to overcome those obstacles. Immune cells like B cells, NK cells, or macrophages are considered targets for immunotherapy, and the regulation of immune-suppressive cells is also part of new immunotherapy. More studies are still warranted, but we foresee these advancements will provide a new paradigm in the immunotherapy of breast cancer.

## Figures and Tables

**Table 1 ijms-26-03920-t001:** Clinical trials for breast cancer using Bispecific antibody.

Drug	Target Antigen	Population	Phase	Trial No.	Status
Ertumaxomab	HER2 and CD3	Low Her2 breast cancer	I	NCT00522457 NCT00452140	Terminated
PF-06671008	P-cadherin and CD3	Advanced solid tumor	I	NCT02659631	Terminated
Tebotelimab	LAG3 and CD3	Unresectable and metastatic malignancy	I	NCT032196268	Completed
ZW25	ECD4 and 2 in Her2	Her2-expressing cancer	I	NCT02892123	Active, not recruiting
D3L-001	Her2/CD47	Her2-positive solid tumor	I	NCT05957536	Recruiting
AK117 (Ivonescimab)	VEFG and PD-1	Metastatic TNBC	II	NCT05227664	Completed
PM8002/BNT327	VEGF and PDL-1	Unresectable, locally advanced or metastatic TNBC	Ib/II	NCT05918133	Recruiting

**Table 2 ijms-26-03920-t002:** Clinical trials for BC using cancer vaccines.

	Target Antigen	Study Population	Phase	NCT	Status
Peptide-based					
Nelipepimut-S	Her2	Operable breast cancer (T1-3N+)	III	NCT01479244	Completed
GP2	Her2	HER2/neu-positive subjects with residual disease or high-risk PCR after both neoadjuvant and postoperative adjuvant trastuzumab-based therapy	III	NCT05232916	Recruiting
AE37	Her2-derived peptide	Node-positive and high-risk node-negative breast cancer patients	II	NCT00524277	Completed
VX-001	hTERT	Advanced breast cancer	I	NCT00573495	Completed
		Metastatic breast cancer	I	NCT01660529	Completed
		Stage III breast cancer	I	NCT00753415	Completed
Protein-based					
	Her2 intracellular domain	Her2-positive breast or ovarian cancer	I		
	Her2	Stage II–III breast cancer	I	NCT00058526	Completed
Carbohydrate-based					
STn-KLH (theratope)	Sialyl-Tn	Metastatic breast cancer	III	NCT00003638	Completed
DNA-based					
PANVAC	CEA and MUC-1	Metastatic breast cancer	I/II	NCT00179309	Completed
	Her2	Metastatic Her2-expressing breast cancer	I	Dnr151:785/2001	Completed
		Stage II-III breast cancer	I/II	NCT00250419 and NCT00647114	Completed
		Advanced-stage Her2-positive breast cancer	I	NCT00436254	Active, not recruiting
Dendritic cell-based					
	Her2	Her2-positive DCIS and early breast cancer	I	NCT02061332	Completed
		Stage II (≥6 + LN), III, or IV breast cancer with >50% HER2 overexpression after adjuvant surgery	I	NCT00005956	Completed
	p53	HLA-A2+ patients with progressive advanced breast cancer	II	Not applicable (Denmark)	Completed
Tumor cell-based (combined with DC)					
		Metastatic breast and renal cancer	I	Not applicable	Completed

**Table 3 ijms-26-03920-t003:** CAR-T therapy in breast cancer.

Target Antigen	Study Population	Phase	NCT	Status
Mesothelin	Mesothelioma, lung and breast cancers	I	NCT02414269	Active, not recruiting
	Metastatic solid tumor	I	NCT03545815	Not applicable
	Metastatic TNBC	I	NCT02792114	Active, not recruiting
C-MET	Metastatic TNBC	I	NCT01837602	Completed
	Metastatic melanoma and TNBC	I	NCT03060356	Terminated
MUC-1	Advanced MUC1-positive breast cancer	I	NCT04020575	Recruiting
ROR-1	Advanced ROR1-positive tumor	I	NCT02706392	Terminated
NKG2D	Colorectal, ovarian, bladder, triple-negative breast, and pancreatic cancers	I	NCT03018405	Not applicable
	Relapsed or refractory solid tumor	I	NCT04107142	Not applicable
Her2	Her2-positive metastatic solid tumor	I	NCT02713984	Withdrawn
	Her2-positive solid tumor	1	NCT04650451	Suspended

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
