# Peer review of "Immunotherapy in Breast Cancer: Beyond Immune Checkpoint Inhibitors"

_ijms, 2025, doi:10.3390/ijms26083920_

Round 1
Reviewer 1 Report
Comments and Suggestions for Authors
The authors present a concise and well-written review of immunotherapies for breast cancer, ranging from checkpoint inhibitors to bispecific antibodies, chimeric antigen receptor T-cell therapies, T-cell receptors, tumor-infiltrating lymphocytes, tumor vaccines, and oncolytic virus therapies.
I have two comments:
1) In the introduction, I suggest mentioning not only that pembrolizumab improved pCR, but also prolonged both event-free and overall survival.
2) In her chapter on tumor vaccines, I suggest also mentioning studies with mRNA vaccines.
Overall, this review provides colleagues who treat breast cancer with a helpful overview of the increasing role of immunotherapies.
Author Response
Reviewer 1:
The authors present a concise and well-written review of immunotherapies for breast cancer, ranging from checkpoint inhibitors to bispecific antibodies, chimeric antigen receptor T-cell therapies, T-cell receptors, tumor-infiltrating lymphocytes, tumor vaccines, and oncolytic virus therapies.
I have two comments:
1) In the introduction, I suggest mentioning not only that pembrolizumab improved pCR, but also prolonged both event-free and overall survival.
2) In her chapter on tumor vaccines, I suggest also mentioning studies with mRNA vaccines.
Overall, this review provides colleagues who treat breast cancer with a helpful overview of the increasing role of immunotherapies.
Response:
- We thank the reviewer’s insightful comments and suggestions. We have added the PFS and OS improved to pembrolizumab in the introduction.
- We updated the tumor vaccine section with recent mRNA vaccines in breast cancer.
Reviewer 2 Report
Comments and Suggestions for Authors
The review article “Immunotherapy in breast cancer; beyond immune checkpoint inhibitors is nice and covers brief overview of many immunotherapeutic strategies for BC’s like BiTE to innate immune modulators. The introduction clearly states what gap this review will cover, which is good but few things can be improved:
- Authors have mostly summarized different therapies. It would be helpful if they include some analysis or discussion about why certain studies that showed promise in preclinical trials didn't succeed later in clinical trials – for example, PF-06671008 (BiTE). Although the original article is cited, adding just a couple of lines to explain why it failed would help readers understand its limitations clearly, similar to how they explained it nicely in section 2.3.1 (2nd paragraph – respiratory failure). Also, section 2.3.3 on TCR gene therapy feels incomplete; maybe briefly mention why targeting NY-ESO-1 or CEA didn’t show success in clinical studies.
- Authors discussed nicely about many therapeis but didn’t include much on radioimmunotherapy , might help to briefly mention whats currently used – mostly beta emitters and if there are new alpha emitters being tried
- Minor but important point – plagiarism report flagged some / lengthy sentences identical with previous papers, these should be rewritten in your words. (Page 5 – ref 77), (Page 7 3rd paragraph), (Page 10 – 2nd para)
- Formatting wise authors should double check and match journal guideline for font, -size, punctuation around references, typos, etc.
adding these points would make the review more complete and useful for readers.
Author Response
The review article “Immunotherapy in breast cancer; beyond immune checkpoint inhibitors is nice and covers brief overview of many immunotherapeutic strategies for BC’s like BiTE to innate immune modulators. The introduction clearly states what gap this review will cover, which is good but few things can be improved:
- Authors have mostly summarized different therapies. It would be helpful if they include some analysis or discussion about why certain studies that showed promise in preclinical trials didn't succeed later in clinical trials – for example, PF-06671008 (BiTE). Although the original article is cited, adding just a couple of lines to explain why it failed would help readers understand its limitations clearly, similar to how they explained it nicely in section 2.3.1 (2nd paragraph – respiratory failure). Also, section 2.3.3 on TCR gene therapy feels incomplete; maybe briefly mention why targeting NY-ESO-1 or CEA didn’t show success in clinical studies.
- Authors discussed nicely about many therapeis but didn’t include much on radioimmunotherapy , might help to briefly mention whats currently used – mostly beta emitters and if there are new alpha emitters being tried
- Minor but important point – plagiarism report flagged some / lengthy sentences identical with previous papers, these should be rewritten in your words. (Page 5 – ref 77), (Page 7 3rd paragraph), (Page 10 – 2nd para)
- Formatting wise authors should double check and match journal guideline for font, -size, punctuation around references, typos, etc.
adding these points would make the review more complete and useful for readers.
Responses: We thank the reviewer’s insightful comments and suggestions. Please see the point by point responses.
- Added our interpretation of the PF-06671008 (BiTE) trial failure: the phase I trial in humans did not show efficacy with significant treatment-related adverse events (TRAE); the most common were cytokine release syndrome (CRS), leading to permanently discontinued treatment in 25% of the patients (NCT 02659631).
For TCR targeting NY-ESO-1 didn’t show success, added additional interpretation. Additional information for CEA TCR therapy in colon and breast cancer trials were added.
- As for radioimmunotherapy, we appreciate the reviewer’s input and agree radiation is certainly an important form of immunotherapy and warrants a separate review due to its complexity. We will not able to add a radioimmune section to the current review.
- All of the sections mentioned were re-written or edited.
- Typos were fixed.